# The Impact of the COVID-19 Pandemic on Healthy Lifestyle Behaviors and Perceived Mental and Physical Health of People Living with Non-Communicable Diseases: An International Cross-Sectional Survey

**DOI:** 10.3390/ijerph19138023

**Published:** 2022-06-30

**Authors:** Salma Azzouzi, Catherine Stratton, Laura Paulina Muñoz-Velasco, Kangxin Wang, Maryam Fourtassi, Bo-Young Hong, Rory Cooper, Joseph K. Balikuddembe, Angela Palomba, Mark Peterson, Uma Pandiyan, Andrei Krassioukov, Deo Rishi Tripathi, Yetsa A. Tuakli-Wosornu, Abderrazak Hajjioui

**Affiliations:** 1Clinical Neuroscience Laboratory, Faculty of Medicine, Pharmacy and Dentistry, University Sidi Mohamed Ben Abdellah, Fez 30050, Morocco; salma.azzouzi1@usmba.ac.ma; 2Department of Physical and Rehabilitation Medicine, University Hospital Hassan II of Fez, Fez 30050, Morocco; 3Department of Chronic Disease and Epidemiology, Yale School of Public Health, New Haven, CT 06510, USA; catherine.stratton@yale.edu (C.S.); kangxin.wang@yale.edu (K.W.); yetsa.tuakli-wosornu@yale.edu (Y.A.T.-W.); 4Amputee Rehabilitation Department, National Institute of Rehabilitation, Mexico City 14389, Mexico; paumv@hotmail.com; 5Laboratory of Life and Health Sciences, Faculty of Medicine of Tangier, Abdelmalek Essaâdi University, Tetouan 90100, Morocco; fourtmary@yahoo.fr; 6Department of Rehabilitation Medicine, St. Vincent’s Hospital, College of Medicine, The Catholic University of Korea, Seoul 06591, Korea; mdhong112@gmail.com; 7Human Engineering Research Laboratories (HERL), US Department of Veteran Affairs, School of Health and Rehabilitation Sciences, University of Pittsburgh, Pittsburgh, PA 15206, USA; rcooper@pitt.edu; 8Department of Disaster Health Sciences, Institute for Disaster Management and Reconstruction, Si Chuan University-Hong Kong Polytechnic University, Chengdu 610207, China; jbalikuddembe.k@scu.edu.cn; 9Department of Medical and Surgical Specialties and Dentistry, University of Campania “Luigi Vanvitelli”, 80138 Naples, Italy; angelapalomba0@gmail.com; 10Department of Physical Medicine and Rehabilitation, Michigan Medicine Neuroscience Graduate Program, University of Michigan, Ann Arbor, MI 48109, USA; mdpeterz@med.umich.edu; 11Qatar Rehabilitation Institute, Hamad Medical Corporation, Doha 3050, Qatar; umapandiyan@gmail.com; 12International Collaboration on Repair Discoveries, Faculty of Medicine, University of British Columbia, Vancouver, BC V6T 1Z4, Canada; andrei.krassioukov@vch.ca; 13Division of Physical Medicine and Rehabilitation, Department of Medicine, University of British Columbia, Vancouver, BC V6T 1Z4, Canada; 14G.F. Strong Rehabilitation Centre, Vancouver, BC V5Z 2G9, Canada; 15Dr. Ram Manohar Lohia Hospital and Post Graduate Institute of Medical Education and Research (PGIMER), New Delhi 110001, India; tripathideorishi@gmail.com

**Keywords:** non-communicable diseases, healthy lifestyle, COVID-19, physical health, mental health

## Abstract

The huge burden and vulnerability imposed by non-communicable diseases (NCDs) during the COVID-19 pandemic highlighted how healthy lifestyle behaviors and the well-being of people living with NCDs need to be prioritized. The aim of our study is to better understand the impact of the COVID-19 pandemic on healthy lifestyle behaviors and perceived mental and physical health among adults living with NCDs, as compared to people without NCDs. We conducted a cross-sectional study using a global online survey through Qualtrics. Over four months, 3550 participants from 65 countries worldwide responded to the survey. The study included 3079 surveys with no missing data (complete survey responses) that were used for analysis. People with NCDs were more likely to report statistically significant worsening physical health (*p* = 0.001) and statistically insignificant worsening mental health (*p* = 0.354) when compared to pre-pandemic levels. They reported lower rates of smoking during the pandemic than those without NCDs, and a statistically significant (*p* < 0.001) relationship was found between weight gain and NCDs. Therefore, the perceived physical and mental health, including changes in body weight and tobacco consumption, of people with NCDs were significantly impacted during the pandemic. In conclusion, this study indicates that the pandemic had a significant impact on perceived physical and mental health, changes in body weight, and tobacco consumption among people with NCDs.

## 1. Introduction

Non-communicable diseases (NCDs) are the leading cause of death globally [1], and one of the major health challenges of the 21st century. While 5.60 million deaths are associated with Coronavirus Disease 2019 (COVID-19) worldwide in the two years since the start of the pandemic in December 2019 [2], NCDs are associated with the deaths of 41 million people each year (approximately 70% of all deaths worldwide) [3]. Four NCDs cause the majority of these deaths: cardiovascular disease, cancer, chronic respiratory diseases, and diabetes [3].

Most NCDs are the result of a combination of genetic, physiological, environmental, and behavioral factors. Healthy lifestyle behaviors (eating a healthy diet, keeping physically active, quitting smoking, and getting enough sleep) are major predictors of mental wellbeing during the COVID-19 pandemic among the general population [4]. Healthy lifestyle behaviors can improve physical and mental health and, therefore, mitigate the burden of NCDs, especially during the COVID-19 crisis [5].

The COVID-19 pandemic had a dramatic impact on healthy lifestyles, mental health, and the quality of life of people worldwide, and presents an unprecedented challenge to public health. The impact of COVID-19 response measures on people with NCDs is multifaceted. Physical distancing or quarantine can lead to poor management of NCD behavioral risk factors, including an unhealthy diet, physical inactivity, and tobacco use [6]. Chronic conditions can also worsen due to stressful situations resulting from restrictions imposed by the pandemic; these stressful situations include insecure economic situations and changes in common health behaviors [6]. 

Public health measures imposed in response to COVID-19 were of great importance in halting the spread of COVID-19, and in protecting people who are especially susceptible to the virus, such as those living with pre-existing NCDs. Notwithstanding, pandemic restrictions also introduced challenges for those living with NCDs, such as disruptions to continuity of care and lifestyle behavior changes. This study aims to assess the impact of the COVID-19 pandemic on the healthy lifestyle behaviors and mental and physical health of adults living with NCDs, as compared to people without NCDs.

## 2. Materials and Methods

### 2.1. The Setting and Participants

This cross-sectional global online survey was conducted between August and December 2020, categorizing countries according to the six World Health Organization (WHO) regions. Participants were recruited using convenience snowball sampling on social media group posts, outpatient clinic distribution, and email blasts targeted to include persons with and without NCDs. Participants were asked to complete an anonymous online questionnaire in one of seven languages, namely, Arabic, Chinese (simplified)**,** English, French, Russian, Spanish, and Korean. Participants were free to withdraw and leave the questionnaire at any time before the final submission of responses.

### 2.2. Survey Questionnaire 

The survey questionnaire (Appendix A) captured socio-demographic characteristics of the participants (e.g., age, gender, country, level of education, and employment status) using unvalidated questions designed by the research team; questions regarding functionality, social engagement, perceived impact of the COVID-19 pandemic on mental and physical health, and healthy lifestyle behavior changes were addressed using two validated instruments: The Washington Group Questions (WGQ) [7] and the International Classification of Functioning, Disability and Health (ICF) [8].

### 2.3. Data Analysis

Analyses were conducted using Statistical Package for the Social Sciences (SPSS) version 26.0 (Faculty of Medicine, Pharmacy and Dentistry, University Sidi Mohamed Ben Abdellah Fez, Morocco) for Windows. Descriptive statistics were used to summarize basic characteristics of participants. The number of participants and percentages (%) were indicated for categorical variables, and means and standard deviations (SD) for continuous variables. The Chi-square test was used to compare the difference among groups with and without NCDs. To assess the impact of the COVID-19 pandemic on healthy lifestyle behaviors and perceived mental and physical health of people with NCDs in our study, univariate logistic regressions were conducted. Results were reported as odds ratios (ORs) with corresponding 95% confidence intervals (CIs), and variables having *p*-value ≤ 0.05 in the univariate analysis were considered statistically significant. As the reference time before and after COVID-19 was a few months, age-related changes were considered negligible. The International Classification of Diseases 11th Revision (ICD11) [9] was used to classify NCDs. Recommendations for healthy eating habits were set according to The Joint WHO/Food and Agriculture Organization of the United Nations (FAO) Expert Consultation on Diet, Nutrition and the Prevention of Chronic Diseases [10].

### 2.4. Ethics

Ethical approval for this study was obtained from the Clinical and Translational Research Oversight Committee of Yale University (IRB# 28723). The study was completed in accordance with the Declaration of Helsinki. All participants gave informed consent before answering the questionnaire, and data confidentiality was respected via anonymous results.

## 3. Results

### 3.1. Socio-Demographic Characteristics of the Study Population

A total of 3550 people around the world responded to this survey; 3079 people (86.73%) from 65 countries who completed the questionnaire were included in the study sample. Another 471 people started the survey, but did not complete their answers; their responses were excluded from the final database. 

The largest proportion of the sample was 25–39 years old (52.8%); participants aged 60 years and over-represented 6.4% of the study sample with a significant correlation between higher age and NCDs. Among respondents, 83.4% were female; an equal distribution was found between men and women regarding NCDs. We noted strong participation from the WHO Region of the Americas, which was also the region with the highest number of confirmed COVID-19 cases. Participants from the Region of the Americas represented 67.3% of the study sample; 52.8% of these participants were living with NCD. Seven percent of participants were from the East Mediterranean Region, and 6.9% were from the Western Pacific Region. Two thirds of participants (63.8%) held postgraduate degrees and only 0.4% had no education or elementary school, with results indicating no statistically significant difference between the two study groups (with and without NCDs). The unemployment rate of our sample was 3.3%, with a statistically significant relationship (*p* < 0.05) between NCDs and unemployment. The survey results revealed that 49.6% of participants (1526) had at least one non-communicable disease. More than 66.3% of participants with NCDs had endocrine, nutritional, or metabolic diseases. Regarding the functional levels of our sample, 16.7% of the study population reported limitations in walking or climbing stairs and 15.6% had hearing and visual impairments. Cognitive impairments (e.g., memory issues or difficulty concentrating) were present in 11.2% of our sample, and 12.0% of participants had some form of communication limitation (e.g., expression or comprehension of language). Difficulties in carrying out activities of daily life (ADLs) (e.g., washing oneself, dressing oneself or eating, drinking, and using the toilet independently) were reported by 4.9% of our population, and difficulties with domestic life activities (shopping, cooking, cleaning house, washing dishes, doing laundry) were reported by 13.6%. Social participation was most restricted in relationships (strangers, family, friends, and colleagues) of 41% of the study participants, and in community, social, and civic life (community, recreation, and religion) of 37.8% of the study participants. The complete demographic findings are reported in Table 1. 

### 3.2. Healthy Lifestyle Behaviors before the COVID-19 Pandemic

Table 2 summarizes the healthy lifestyle behaviors of the study population before the COVID-19 pandemic. Before the COVID-19 pandemic, only 20.9% of participants engaged in at least 150 min per week of physical activity as recommended by the WHO; 16.8% of the study population had at least seven uninterrupted hours per day of sedentary time. A statistically significant relationship was found between the presence of NCDs and physical activity (*p* < 0.001) and sedentary time (*p* < 0.001). Results noted in Table 2 also reveal that 43.2% of participants consumed five servings of fruits and vegetables at least five days per week, and 10.9% were smokers.

### 3.3. General Impact of COVID-19 Pandemic

Almost all participants (98.0%) in our study confirmed that the COVID-19 crisis changed their daily life as a result of requirements to maintain social distancing (87.9%) and wear a mask in public spaces (87.0%), and experiencing lockdown(s) (“shelter in place”) (53.2%). There was a statistically significant relationship (*p* < 0.05) between the impact of the COVID-19 pandemic on daily life and the presence of NCDs (Table 3).

### 3.4. Healthy Lifestyle Behaviors during the COVID-19 Pandemic 

Participants’ lifestyle behaviors (physical activity, sedentary behaviors, eating habits, sleeping habits, weight changes, and smoking) during the COVID-19 pandemic are presented in Table 3. Regarding physical activity, only 20.9% of participants engaged in at least 150 min per week of physical activity as recommended by the WHO before the pandemic; 51.8% of participants reported a decrease in physical activity level during the COVID-19 pandemic. The difference between the two groups was not statistically significant.

Changes were noted in participants’ uninterrupted sedentary time, such as sitting, before and during the pandemic. Before the COVID-19 pandemic, approximately 36.0% of participants with NCDs reported 0–3 h of sedentary activity, nearly half of respondents (44.9%) reported 1–4 h of sedentariness, and 19.7% reported they spent more than four hours being sedentary. Surprisingly, the pandemic had a positive impact on the sedentary behavior of participants regardless of the presence or absence of NCDs. There was a 20.5% increase in the number of non-sedentary people with or without NCDs. The difference between the two groups was not statistically significant (Table 2).

Sleep disturbance was also studied in the survey. Results show that 1312 (42.6%) participants slept at least seven hours per night more than four days per week before the pandemic, which is consistent with WHO recommendations. Regarding sleep patterns during the COVID-19 pandemic, 1174 (38.1%) participants noted they slept less than before, and 1230 (39.9%) respondents’ sleep habits remained the same.

The survey investigated variations in fruit and vegetable intake before the pandemic; 1329 respondents (43.2%) ate the recommended minimum five servings of fruit and vegetables per day. Regarding eating habits during the pandemic, 1614 (52.4%) participants reported that their fruit and vegetable intake remained unchanged, and 619 (20.1%) reported eating less fruit and vegetables than before.

Regarding tobacco use before the COVID-19 pandemic, 89.1% of study participants indicated they were non-smokers, 49.2% of whom had at least one NCD. During the COVID-19 pandemic, only 2% of non-smokers started smoking. On the other hand, 10.9% of participants reported that they were tobacco users, 52.7% of whom had at least one NCD. Interestingly, during the COVID-19 outbreak 5.6% of participants and 42.6% of smokers reported a reduction in tobacco use; however, 4.7% of participants and 26.2% of smokers reported increased tobacco use.

Perceived weight gain during the pandemic was reported by 45.3% of participants, of whom 49.5% had NCDs. A statistically significant relationship (*p* < 0.001) was found between weight gain and NCDs (Table 3).

### 3.5. Impact of COVID-19 Pandemic on Physical and Mental Health

Regarding the impact of COVID-19 on physical and mental health, two-thirds (67.8%) of participants reported deterioration in their self-perceived mental health during the COVID-19 pandemic. Of these participants, 48.5% confirmed the negative impact of the pandemic on their self-perceived physical health with regard to the presence of NCD, which demonstrates a statistically significant (*p* = 0.001) association between the deterioration of physical health in people living with NCD and the COVID-19 pandemic (Table 3).

### 3.6. Lifestyle during the COVID-19 Pandemic: Univariate Logistic Regression Analysis

Table 4 shows the results of the univariate logistic regression analysis, as well as the corresponding ORs with 95% CIs of the risks of lifestyle changes during the COVID-19 pandemic associated with the presence of NCDs. 

The results reveal that the presence of NCDs was significantly associated with 24% greater odds of a positive change in daily fruit and vegetable intake (OR = 1.24, 95% CI: 1.06–1.45; *p* = 0.008), and 46% greater odds of reduced tobacco use (OR = 1.46, 95% CI: 1.06–1.99; *p* = 0.017). Conversely, the presence of NCDs was negatively and significantly associated with 27% higher odds of reducing night-time sleep hours (OR = 1.27, 95% CI: 1.10–1.47; *p* = 0.001), and 27% higher odds of reporting a deterioration in physical health (OR = 1.27, 95% CI: 1.10–1.46; *p* = 0.001). Finally, changes in PA levels, sedentary behaviors, and perceived mental health status were not significantly associated with the presence of NCDs.

## 4. Discussion

To our knowledge, the present survey is one of only a few that evaluate the impact of the COVID-19 pandemic on lifestyle behaviors and the perceived physical and mental health of people living with NCDs. This study is especially unique as it gathered data from 65 countries worldwide, which had different policies and responses during the ongoing COVID-19 crisis.

Our findings show that the pandemic effected changes in lifestyle and the perceived physical and mental health of people living with NCDs. The most striking positive changes were described in the self-reported sedentariness and tobacco use of people with NCDs. However, the negative changes exceed the positive ones. The main negative changes were in self-reported physical activities, weight gain, and the self-perceived physical and mental health of people living with NCDs.

The first finding of our study is that the pandemic had a different impact on self-reported sedentary behaviors compared to physical activities in people living with NCDs. The decrease in physical activity during the pandemic can be partially attributed to the closure of gyms, swimming pools, and exercise clubs, in addition to laws limiting access to outdoor spaces and freedom of movement restrictions, all of which inevitably reduced opportunities to exercise [11]. This was potentially exacerbated by a lack of motivation to continue physical activity due to depression or anxiety generated by social isolation [12]. This is of particular importance to people with NCDs, for whom physical activity is essential in the control of symptoms and associated risk factors for health complications such as obesity, hypertension, and elevated glucose levels [13]. Conversely, our study identified a decrease in sedentary behaviors in people living with NCDs during the pandemic, most likely due to the fact that the majority of participants recruited were employed (82%) and had a high level of education. One possible explanation is that participants with higher education levels may have more control over the way they work, and may use skills that enable them to engage in healthy behaviors [14]. This agrees with the data in the literature showing that a higher education level and employment are associated with lower leisure-time sedentary behavior [14,15,16]. Moreover, it is important to understand that people who are “physically inactive” (do not meet physical activity recommendations for their age) are not necessarily defined as “sedentary” (engaging in sedentary behaviors such as sitting or leaning with an energy expenditure of 1.5 metabolic equivalent task (MET) or less), since physical inactivity and sedentary behaviors are two different concepts that are commonly confused. Therefore, a person can be physically inactive and not sedentary at the same time [17].

Our study also illustrates the negative impact of the pandemic on body weight. The present study findings are in line with a study carried out in Addis Ababa indicating that for people with NCDs in their study population, a decrease in physical activity during the COVID-19 pandemic was associated with weight gain [18]. Another study from Europe shows that restrictive measures during the COVID-19 pandemic created challenges to maintaining appropriate levels of physical activity and a healthy weight. Reduced physical activity, including work commutes, exercise, and sports for recreational purposes, possibly led to an increase in obesity [6].

This study’s finding of reported changes in tobacco use is in line with a study carried out in Bangladesh that also found that participants who had pre-existing NCDs had a lower chance of increased tobacco use during the pandemic [19]. This is potentially because those living with NCDs are more concerned about the deadliest effects of COVID-19, namely, its respiratory effects; therefore, they lowered their tobacco use [19]. 

The perceived worsening of physical and mental health identified here is in keeping with other studies which demonstrate the adverse impact of the COVID-19 pandemic on the mental health of most patients with NCDs (specifically, increased feelings of stress and loneliness) [20]. Reduced physical activity levels and physical fitness were also identified as serious concerns during home confinement resulting from the pandemic [21]. This demonstrates a rapidly expanding need for practical recommendations and campaigns to promote an active lifestyle to mitigate unwanted physical and social confinement consequences during pandemics, especially for people with NCDs. Some countries formed national media campaigns. This approach is being implemented in some developed countries, including Australia and the United Kingdom. These countries invested in national campaigns on TV, radio channels, and social media to advocate for the importance of physical activity to maintain people’s health during the COVID-19 pandemic [22]. The positive impact of such a media campaign is evident in a recent study conducted in England; nearly two thirds of adults reported considering exercise to be more important than ever during the current COVID-19 outbreak [23]. In addition, approximately 65% of people believed that exercise helped them maintain their mental health during the pandemic [23].

Additionally, unprecedented access to online digital content during the COVID-19 pandemic enabled the world to maintain connectedness to their recreational and fitness communities, while also accessing direct health and fitness-related programs [24]. Digital health technologies include technical solutions to promote self-care for people with NCDs who cannot access services, and can be implemented individually in cases of isolation, home-confinement, and social distancing. Being a member of a virtual fitness program, for instance, increases the level of PA, and reduces loneliness and social isolation [25]. Mobile health apps may also help promote active living during COVID-19. The positive effect of these apps on improving physical activity is reported in some studies [26,27]. 

Taken together, these findings suggest that the COVID-19 pandemic significantly impacted the lifestyle of people with NCDs, revealing how vulnerable they are. This underscores the importance of strong health systems giving more priority to people living with NCDs whose needs for care and treatment can be challenging. According to the WHO survey conducted in June 2020 [28], prevention and treatment services for NCDs have been severely disrupted since the COVID-19 pandemic began. Rehabilitation services, hypertension management, and diabetes and diabetic complications management were the three main services disrupted by the COVID-19 crisis [28]. One interesting example of maintaining care delivery to people with NCDs during the pandemic is the Field Epidemiology Training Program (FETP) NCD track in India; program fellows helped local public health departments during the pandemic and maintained important NCD services for people with limited access to medical services and at higher risk for severe COVID-19 outcomes. They even introduced a telemedicine feature which consists of a mobile-based application used to track follow-up of NCD patients; they also helped community health workers using the app to connect people needing NCD services with local doctors [29]. The India FETP NCD program started in 2018. Is fellows are trained to fight both NCDs and infectious diseases, and continued to do so when the pandemic disrupted regular health services. Governments and medical and health institutions at various levels should develop similar NCD care programs, and adapt telehealth approaches and digital health care solutions to maintain the continuum of care for those with NCDs both during and after the pandemic.

The COVID-19 pandemic demonstrated that people living with NCDs are at higher risk, and the most vulnerable, during health crises. National healthcare systems were unable to fully protect population health by implementing programs focused on prevention, early diagnosis, screening, treatment, and rehabilitation for NCDs. The ongoing COVID-19 pandemic highlights the urgent need to empower people affected by NCDs and to tackle the growing public health burden imposed by NCDs on health resources. It should act as a catalyst for governments to make investments that enable people with NCDs to improve their health-related behaviors.

To fully understand the consequences of the pandemic, we need to assess its overall impact on the incidence, management, and outcomes of chronic disease. More methodological research is required to better understand if the incidence and prevalence of chronic diseases increased because of pandemic-related health behaviors or other challenges. In addition, further long-term studies and longitudinal surveillance are needed to suggest ways to improve the healthy lifestyles and wellbeing of people living with NCDs during pandemics.

## 5. Strengths and Limitations

Strengths of this study include the large survey sample size, which was conducted in 58 countries in all six WHO regions. Further, the survey gathered data on multiple self-reported lifestyle behavior metrics and self-perceived physical and mental well-being. Nevertheless, for a more accurate interpretation of the results, several limitations should be considered. First, despite the international dimension of the study, the sample size was not representative of the potential population of people with NCDs or without NCDs in different regions across the 65 countries from which the sample was drawn. Moreover, the convenience snowball sampling strategy that we employed does not allow us to calculate the sample size, and can have a potential sampling bias and margin of error, because participants refer people they know who have similar traits. Even after referral, some people declined to participate in the research study, resulting in a low response rate and inconclusive results. Second, due to the cross-sectional nature of our sampling strategy, inference on causality is not possible. A longitudinal study with multiple sampling opportunities from populations with NCDs and without NCDs may provide a better opportunity to observe changes in individuals’ health behaviors and perceived physical and mental health during the COVID-19 pandemic. Third, the use of subjective questions in the survey questionnaire without providing further details (e.g., asking about visual and hearing impairments without giving accurate definitions, using a Yes/No question to ask participants if they live with one or more NCDs, asking questions about health and lifestyle factors pre-pandemic, suggesting recall bias, etc.) was another limitation. Clear and unambiguous questions are prerequisites for obtaining reliable and valid information that can be analyzed statistically [30]. Finally, our sampling method relied heavily on technology and Internet connectivity; therefore, we were only able to collect responses from participants with Internet access and a certain level of technology proficiency. Additionally, we noted that our sample mainly comprised young women with a high education level, and was potentially not representative of the global population.

## 6. Conclusions

In conclusion, the results of our study show that the COVID-19 pandemic is negatively impacting self-perceived physical and mental health, particularly among people living with NCDs. Interestingly, the COVID-19 pandemic appears to be having a positive impact on self-reported sedentary behaviors and tobacco use in people with NCDs. Public health interventions are needed to address healthy lifestyle behaviors during and after the COVID-19 pandemic. 

## Figures and Tables

**Table 1 ijerph-19-08023-t001:** Demographic data of survey respondents.

		Total N (%) 3079 (100.0)	NCDs N (%) 1526 (49.6)	No NCD N (%) 1553 (50.4)	*p*-Value	
** *Age group (y)* **						
	18 to 24	76 (2.5)	17 (1.1)	59 (3.8)	<0.001 *	
	25 to 39	1625 (52.8)	727 (47.6)	898 (57.8)		
	40 to 59	1179 (38.3)	652 (42.7)	527 (33.9)		
	60 and above	196 (6.4)	128 (8.4)	68 (4.4)		
	No response	3 (0.1)	2 (0.1)	1 (0.1)		
** *Gender* **						
	Women	2554 (82.9)	1272 (83.4)	1282 (82.5)	0.889	
	Men	514 (16.7)	248 (16.3)	266 (17.1)		
	Other/No response	11 (0.4)	6 (0.4)	5 (0.3)		
** *WHO Region of Origin* **						
	The Region of the Americas (AMRO)	2073 (67.3)	1094 (52.8)	979 (47.2)		
	The European Region (EURO)	60 (1.9)	18 (30.0)	42 (70.0)		
	The Eastern Mediterranean Region (EMRO)	217 (7)	75 (34.6)	142 (65.4)		
	The African Region (AFRO)	24 (0.8)	10 (41.7)	14 (58.3)		
	The South-East Asia Region (SEARO)	49 (1.6)	26 (53.1)	23 (46.9)		
	The Western Pacific Region (WPRO)	211 (6.9)	86 (40.8)	125 (59.2)		
	Region not specified	445 (14.5)	217 (48.8)	228 (51.2)		
** *Education Level* **						
	No school	5 (0.2)	3 (0.2)	2 (0.1)	0.653	
	Primary/Elementary	7 (0.2)	3 (0.2)	4 (0.3)		
	Secondary/High School	110 (3.6)	63 (4.1)	47 (3)		
	Bachelor or equivalent	988 (32.1)	485 (31.8)	503 (32.4)		
	Postgraduate	1964 (63.8)	969 (63.5)	995 (64.1)		
	Other/No response	5 (0.2)	3 (0.2)	2 (0.1)		
*Employment Status*						
	Employed (part-time or full-time)	2535 (82.3)	1240 (81.3)	1295 (83.4)	<0.001 *	
	Homemaker	149 (4.8)	71 (4.7)	78 (5)		
	Retired	148 (4.8)	106 (6.9)	42 (2.7)		
	Student	137 (4.4)	47 (3.1)	90 (5.8)		
	Unemployed	103 (3.3)	58 (3.8)	45 (2.9)		
	Other/No response	7 (0.2)	4 (0.3)	3 (0.2)		
*Number of NCD*						
	0	1553 (50.4)	0 (0)	1553 (100)	<0.001 *	
	1	1254 (40.7)	1254 (82.2)	0 (0)		
	2	244 (7.9)	244 (16)	0 (0)		
	≥3	28 (0.9)	28 (1.8)	0 (0)		
** *NCD category (ICD11)* **						
	05 Endocrine, nutritional or metabolic diseases	1011 (32.2)	1011 (66.3)	0 (0)	<0.001 *	
	11 Diseases of the circulatory system	331 (10.8)	331 (21.7)	0 (0)		
	12 Diseases of the respiratory system	239 (7.8)	239 (15.7)	0 (0)		
	08 Diseases of the nervous system	72 (2.3)	72 (4.7)	0 (0)		
	02 Neoplasms	35 (1.1)	35 (3.2)	0 (0)		
	Other diseases	69 (2.2)	69 (4.5)	0 (0)		
** *Functioning levels and social participation restriction* **						
	Seeing/Hearing	479 (15.6)	289 (18.9)	190 (18.6)	<0.001 *	
	Walking or climbing steps	514 (16.7)	339 (22.21)	175 (11.3)	<0.001 *	
	Remembering or concentrating (cognitive difficulty)	345 (11.2)	164 (10.7)	145 (9.3)	0.01 *	
	Self-care (washing oneself, using the toilet, dressing, eating, and drinking)	149 (4.9)	99 (6.5)	50 (3.2)	<0.001 *	
	Communicating (understanding or being understood)	371 (12.0)	202 (13.2)	169 (10.9)	0.13	
	Domestic life (shopping, cooking, cleaning house, washing dishes, and doing laundry)	420 (13.6)	247 (16.2)	173 (11.1)	<0.001 *	
	Community, social, and civic life (community, recreation, and religion)	1164 (37.8)	638 (41.8)	526 (33.9)	<0.001 *	
	Relationships (strangers, family, friends, and colleagues)	1262 (41.0)	678 (44.4)	584 (37.6)	0.004 *	

* statistically significant.

**Table 2 ijerph-19-08023-t002:** Healthy lifestyle behaviors before the COVID-19 pandemic in people with and without NCDs.

Lifestyle before COVID-19 Pandemic	Total N (%) 3079 (100.)	NCDs N(%) 1526 (49.60)	No NCD N(%) 1553 (50.4)	*p*-Value
** *At least 30 min of moderate physical activity every day (Number of days per week)* **				
0	632 (20.4)	349 (22.9)	283 (18.2)	0.001 *
1	575 (18.7)	303 (19.9)	272 (17.5)	
2–4	1228 (39.9)	580 (38)	648 (41.7)	
≥5	644 (20.9)	294 (19.3)	350 (22.5)	
** *Sedentary Time (Number of hours per day)* **				
≤1	192 (6.2)	92 (6.0)	100 (6.4)	<0.001 *
>1–3	1003 (32.6)	452 (29.6)	551 (35.5)	
>3–7	1367 (44.4)	682 (44.7)	685 (44.1)	
≥ 7	517 (16.8)	300 (19.7)	217 (14.0)	
** *≥7 h of sleep per night (Number of days per week)* **				
0	224 (7.3)	125 (8.2)	99 (6.4)	0.079
1	421 (13.7)	215 (14.1)	206 (13.3)	
2–4	1122 (36.4)	553 (36.2)	569 (36.6)	
≥5	1312 (42.6)	633 (41.5)	679 (43.7)	
** *≥5 servings of fruits and vegetables per day (Number of days per week)* **				
0	126 (4.1)	69 (4.5)	57 (3.7)	0.203
1	323 (10.5)	172 (11.3)	151 (9.7)	
2–4	1301 (42.3)	636 (41.7)	665 (42.8)	
≥5	1329 (43.2)	649 (42.5)	680 (43.8)	
** *Tobacco use* **				
Yes	336 (10.9)	177 (11.6)	159 (10.2)	0.478
No	2743 (89.1)	1349 (88.4)	1394 (89.8)	

* statistically significant.

**Table 3 ijerph-19-08023-t003:** Daily life changes and healthy lifestyle behaviors in people with and without NCD during the COVID-19 pandemic.

Lifestyle during COVID-19 Pandemic		Total N (%) 3079 (100.0)	NCDs N (%) 1526 (49.6)	No NCD N (%) 1553 (50.4)	*p*-Value
** *Impact of COVID-19 pandemic* **					
	General impact	3016 (98.0)	1502 (98.4)	1514 (97.5)	0.043 *
	Maintained social distancing	2705 (87.9)	1347 (88.3)	1358 (87.4)	0.26
	Wore mask in public	2680 (87.0)	1349 (88.4)	1331 (85.7)	0.015 *
	Experienced lockdown (shelter in place)	1637 (53.2)	843 (55.2)	794 (51.1)	0.012 *
	Income loss/reduction	1039 (33.7)	545 (35.7)	494 (31.8)	0.012 *
	Experienced total job loss/unemployment	162 (5.3)	91 (6)	71 (4.6)	0.50
	Other impacts	229 (7.4)	129 (8.5)	100 (6.4)	0.020 *
** *Daily physical activity was* **					
	Less than before	1595 (51.8)	781 (51.2)	814 (52.4)	0.923
	Same as before	885 (23.4)	445 (29.1)	440 (28.3)	
	More than before	599 (19.5)	300 (19.7)	299 (19.3)	
** *Sedentary time was* **					
	Less than before	1211 (39.3)	626 (41.0)	585 (37.7)	0.923
	Same as before	1355 (44.0)	652 (42.7)	703 (45.3)	
	More than before	513 (16.7)	248 (16.3)	265 (17.0)	
** *Nightly hours of sleep were* **					
	Less than before	1174 (38.1)	625 (41.0)	549 (35.4)	0.009 *
	Same as before	1230 (39.9)	573 (37.6)	657 (42.3)	
	More than before	675 (21.9)	328 (21.5)	347 (22.3)	
** *Daily consumption of fruits and vegetables( ≥ 5 servings) was* **					
	Less than before	619 (20.1)	318 (20.8)	301 (19.4)	0.013 *
	Same as before	1614 (52.4)	756 (49.5)	858 (55.2)	
	More than before	846 (27.5)	452 (29.6)	394 (25.4)	
** *Daily eating habits were* **					
	Worse than before	939 (30.5)	506 (33.2)	433 (27.9)	0.001 *
	Same as before	1176 (38.2)	533 (34.9)	643 (41.5)	
	Better than before	964 (31.3)	487 (31.9)	477 (30.7)	
** *Tobacco use was* **					
	Less than before	173 (5.6)	101 (6.6)	72 (4.6)	0.089
	Same as before	2762 (89.7)	1355 (88.8)	1407 (90.6)	
	More than before	144 (4.7)	70 (4.6)	74 (4.8)	
** *Physical health and physical fitness were* **					
	Worse than before	1401 (45.5)	740 (48.5)	661 (42.6)	0.001 *
	Same as before	1132 (36.76)	510 (33.4)	622 (40.0)	
	Better than before	546 (17.7)	276 (18.1)	270 (17.4)	
** *Mental health and emotional wellness were* **					
	Worse than before	2050 (66.6)	1034 (67.8)	1016 (65.4)	0.354
	Same as before	819 (26.6)	386 (25.3)	433 (27.9)	
	Better than before	210 (6.8)	106 (6.9)	104 (6.7)	
** *Weight Loss* **					
	Yes	824 (26.8)	410 (26.9)	414 (26.7)	0.800
	No	2105 (68.4)	1036 (67.9)	1069 (68.8)	
	I do not know	150 (4.9)	80 (5.3)	70 (4.5)	
** *Weight gain* **					
	Yes	1397 (45.3)	755 (49.5)	642 (41.3)	<0.001 *
	No	1538 (50.0)	704 (46.1)	834 (53.7)	
	I do not know	144 (4.7)	67 (4.4)	77 (5.0)	
** *Weight gain during lockdown (kg)* **					
	<2.5	458 (14.9)	183 (12.0)	275 (17.7)	<0.001 *
	2.5–4.99	716 (23.3)	411 (26.9)	305 (19.6)	
	5–9.99	177 (5.7)	121 (7.9)	56 (3.6)	
	≥10	42 (1.4)	38 (2.5)	4 (0.3)	
	I do not know	1686 (54.8)	773 (50.7)	913 (58.8)	

* statistically significant.

**Table 4 ijerph-19-08023-t004:** Association between NCDs and healthy lifestyle behaviors and perceived mental and physical health during the COVID-19 pandemic (Predicted Probability) using univariate logistic regression analysis.

		Univariate Logistic Regression OR (95% CI)	p-Value
** *Level of PA* **			
	Reduced	1.05 (0.91–1.21)	0.493
	Increased	1.02 (0.85–1.22)	0.776
	Same level	1.04 (0.89–1.21)	0.611
** *Daily fruit and vegetable intake* **			
	Improved	1.24 (1.06–1.45)	0.008 *
	Worsened	1.09 (0.92–1.30)	0.313
	Same daily consumption	0.79 (0.69–0.92)	0.002 *
** *Hours of night time sleep* **			
	Increased	0.95 (0.80–1.12)	0.569
	Reduced	1.27 (1.10–1.47)	0.001 *
	Same hours	0.82 (0.71–0.95)	0.007 *
** *Tobacco use* **			
	Reduced	1.46 (1.06–1.99)	0.017 *
	Increased	0.96 (0.69–1.34)	0.815
	Same tobacco use	0.82 (0.65–1.04)	0.100
** *Sedentary behaviors* **			
	Improved	1.15 (1.00–1.33)	0.057
	Worsened	0.94 (0.78–1.14)	0.545
	Same behaviors	0.90 (0.78–1.04)	0.156
** *Perceived physical health* **			
	Improved	1.05 (0.87–1.26)	0.611
	Deteriorated	1.27 (1.10–1.46)	0.001 *
	Same perceived physical health	0.75 (0.65–0.87)	<0.001 *
** *Perceived mental health* **			
	Improved	1.04 (0.79–1.38)	0.780
	Deteriorated	1.11 (0.96–1.29)	0.169
	Same perceived mental health	0.88 (0.75–1.03)	0.104

OR: odds ratio; CI: confidence interval; NCDs: non-communicable diseases; PA: physical activity. Meaning of OR: OR is a measure of association between the presence of NCDs (as exposure or independent variables, or risk factors and confounders) and healthy lifestyle behaviors and perceived mental and physical health during the COVID-19 pandemic (as the outcome variable or dependent variable of this logistic regression). OR = 1: NCDS does not affect odds of outcome. OR > 1: NCDs associated with higher odds of outcome. OR < 1: NCDs associated with lower odds of outcome. * statistically significant.

## Data Availability

Data supporting reported results can be obtained on request from the authors.

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
