# Peer review of "The Impact of the COVID-19 Pandemic on Healthy Lifestyle Behaviors and Perceived Mental and Physical Health of People Living with Non-Communicable Diseases: An International Cross-Sectional Survey"

_ijerph, 2022, doi:10.3390/ijerph19138023_

Round 1

Reviewer 1 Report

First of all, I would like to congratulate the authors of the article, they wrote an interesting article which shows the influence of COVID 19 pandemic on the habits and health related variables in 3550 participants, making a distinction between NCDs and non NCDs.

Minor changes:

Please check the results data and add decimals in the following lines.

Line 126 52.8%.

Line 130 66.3%

Line 131 83.4%

Lines 178-186: Indicate de table where the data were obtained, it is not clear.

Discretionary Revisions (which are recommendations for improvement but which the author can choose to ignore)

I recommend describing the results in the order of appearance in the tables

Although causality cannot be analyzed, Prevalence Ratios can be calculated. I recommend performing more precise analyses that help to know which variables can be potential confounders.

Author Response

Response to Reviewer 1 Comments

Point 1: First of all, I would like to congratulate the authors of the article, they wrote an interesting article which shows the influence of COVID 19 pandemic on the habits and health related variables in 3550 participants, making a distinction between NCDs and non NCDs.

Response 1: We thank the Reviewer for this positive feedback. We would also like to thank the reviewer very much for the careful reading and the many good suggestions, which certainly contribute to the quality of the manuscript.

Point 2: Minor changes: Please check the results data and add decimals in the following lines.

Line 126 52.8%. (Now lines 142 and 148)

Line 130 66.3% (Now line 155)

Line 131 83.4% (Now line 144)

Lines 178-186: Indicate de table where the data were obtained, it is not clear. (Now line 380).

Response 2: Thank you. We have corrected the decimals and specified the table according to your suggestion.

Point 3: I recommend describing the results in the order of appearance in the tables

Response 3: Thank you, we have rearranged the findings according to your suggestions.

Point 4: Although causality cannot be analyzed, Prevalence Ratios can be calculated. I recommend performing more precise analyses that help to know which variables can be potential confounders.

Response 4: Thank you for raising this issue and which did other reviewers also mention. We used a logisticregression to calculate odds risk in order to assess the risk of impacting healthy lifestyle inpeople with NCDs (see Table 4).

Although prevalence Ratios can be calculated, we have deemedit more relevant in this paper to use OR and logistic regression.

We are of course open if you consider an additional statistical review as needed, but hope that our response is able to address your concerns satisfactorily.

Reviewer 2 Report

The paper is rather interesting and informative. There are some problems with formulations that I mention later; but the major problem I see is lack of consideration of limitations' obedience as a factor affecting the health parameters. Since the authors obtained a considerable sample of non-abiding respondents - above 10% according to Table 3 - it would be extremely useful to analyze the association of obedience with self-perceived health.

There additional issues that I want to point on.

First, regarding the Abstract: "People with NCDs were more likely to report worsening physical and mental health... as indicated by a statistically significant finding in physical health (p=0.001), but no statistically significant result regarding mental health (p=0.354)." This sentence should be reformulated since no worsening in mental health was observed.

line 58: "While Coronavirus Disease 2019 (COVID-19) killed 5.60 million people worldwide..." - should be reformulated since most COVID-positive deceased had multiple comorbidities. E.g.: "While Coronavirus Disease 2019 (COVID-19) was associated with 5.60 million people worldwide..."

line 77: "Halting the spread of COVID-19 is of great importance" - no connection with the topic of this paper; effectiveness of lockdowns etc. in "halting" COVID-19 is a separate and non-trivial issue.

line 310: ??? - Should we add this paragraph “implications for public health” at the end of the discussion? :"

Author Response

Response to Reviewer 2 Comments

Point 1: The paper is rather interesting and informative. There are some problems with formulations that I mention later; but the major problem I see is lack of consideration of limitations' obedience as a factor affecting the health parameters. Since the authors obtained a considerable sample of non-abiding respondents - above 10% according to Table 3 - it would be extremely useful to analyze the association of obedience with self-perceived health.

Response 1: Thank you for this valuable suggestion, which we included now in the table 4. We used logistic regression to analyze the association between the presence of NCDs and a change in healthy lifestyle.We hope that our response is able to address your concerns satisfactorily and we are of course open if you consider an additional statistical review as needed

Point 2:There additional issues that I want to point on. First, regarding the Abstract: "People with NCDs were more likely to report worsening physical and mental health... as indicated by a statistically significant finding in physical health (p=0.001), but no statistically significant result regarding mental health (p=0.354)." This sentence should be reformulated since no worsening in mental health was observed.

Response 2: We changed those sentences according to your good suggestions. Thank you.

Point 3: Line 58: "While Coronavirus Disease 2019 (COVID-19) killed 5.60 million people worldwide..." - should be reformulated since most COVID-positive deceased had multiple comorbidities. E.g.: "While Coronavirus Disease 2019 (COVID-19) was associated with 5.60 million people worldwide..."

Response 3: We have changed the sentence according to your suggestion, thank you. (now line 64).

Point 4: Line 77: "Halting the spread of COVID-19 is of great importance" - no connection with the topic of this paper; effectiveness of lockdowns etc. in "halting" COVID-19 is a separate and non-trivial issue.

Response 4:Thank you for your good suggestion. We have rephrased this to connect the ideas about the connection between public health measure to halt the spread of COVID-19 and effects of healthy lifestyle factors(Now line 83).

Point 5: Line 310: ??? - Should we add this paragraph “implications for public health” at the end of the discussion? :"

Response 5: We dropped this part of the sentence and we corrected this error, thank you for detecting it.

Reviewer 3 Report

The topic of the manuscript is interesting since speaks about lifestyle changes due to covid-19 pandemic.

The introduction provide enough information. 

The methods are well written, however the statistical methods is poor. Actually, the authors tested the differences using a chi square test, that it is correct, but considering the large sample size, statistical significance could happen because of chance. For this reason, I would suggest to use a logistic regression in order to assess odds risk of, as for instance, more sedentary time during/after covid-19 pandemic or comparing those with and without NCDs. If possible, please also consider statistical adjustment for the main confounding as sex, age, and socioeconomic status.

Did the authors used a validated questionnaire? If no please state. if yes please add validation data (also as supplementary material if needed).

A graphical abstract could increase the interest of the readers.

lines 310-311 should be removed

for transparency, the questionnaire should be attached as supplementary materials.

Author Response

Response to Reviewer 3 Comments

Point 1: The topic of the manuscript is interesting since speaks about lifestyle changes due to covid-19 pandemic. The introduction provides enough information. 

Response 1:Thank you very much for this positive feedback and for the careful reading and the many good suggestions, which certainly contribute to the quality of the manuscript.

Point 2: The methods are well written, however the statistical methods is poor. Actually, the authors tested the differences using a chi square test, that it is correct, but considering the large sample size, statistical significance could happen because of chance. For this reason, I would suggest to use a logistic regression in order to assess odds risk of, as for instance, more sedentary time during/after covid-19 pandemic or comparing those with and without NCDs. If possible, please also consider statistical adjustment for the main confounding as sex, age, and socioeconomic status.

Response 2:Thank you for this valuable suggestion, which certainly contributes to the quality of the manuscript. We used a logistic regression to measure the Odds ratio and to look for the risks of modification of lifestyle and mental and physical health linked to the presence of chronic diseases (see Table N ° 4).

 We now give more additional statistical details that help to help readers interpret the results. On the other hand, we are aiming to compare two groups (with and without NCD, regarding the life style change during covid pandemic, and not exploring risk factors to developing NCDs, which were present within participants before the pandemic. Consequently, multivariate regression wouldn’t be suitable as no risk factors are explored. We are of course open if youconsider an additional statistical review as needed, but hope that our response is able toaddress your concerns satisfactorily.

Point 3:Did the authors used a validated questionnaire? If no please state. if yes please add validation data (also as supplementary material if needed).

Response 3: Thank you for this valuable suggestion. We specify which parts of the survey were validated andwhich were not validated (Now line 103).

Point 4: A graphical abstract could increase the interest of the readers.

Response 4: Thank you for this valuable suggestion. We have added a graphical abstract to increase the interest of the readers

Point 5: Lines 310-311 should be removed

Response 5:Thank you. This part of the sentence has been removed.

Point 6: For transparency, the questionnaire should be attached as supplementary materials.

Response 6:Thank you for this valuable suggestion. We have added this as Supplementary Material.

Round 2

Reviewer 2 Report

Table 4: the table legend must clearly state:

what was the independent variable of the regression

what is OR

what is the meaning of OR>1 (more or less for people with NCD?)

Also:

Level of PA - Level of Physical Activity? "PA" is not explained

All P-values should be with the same precision. E.g., 0.78 => 0.780. If p<0.0005, just write 0.000

Author Response

Dear Reviewer,

We are truly grateful that you give us the opportunity to revise the current version of our manuscript for publication in the IJERPH, thank you very much.

We also thank the reviewers for taking the time to provide us with helpful comments to further improve our manuscript.

Please find a point-to-point reply to all comments below. The revision made is highlighted in yellow.

Yours sincerely,

The authors

Point 1: Table 4: the table legend must clearly state:

  • What was the independent variable of the regression?
  • What is OR?
  • What is the meaning of OR>1 (more or less for people with NCD?)

Response 1: We’ve edited the table legend according to your good suggestions. Thank you for pointing this out and for the careful reading and the many good suggestions, which certainly contribute to the quality of the manuscript.

1) The outcome variable or dependent variable of the logistic regression is the Healthy Lifestyle Behaviors, Perceived Mental and Physical Health during the COVID-19 pandemic. The independent variables or risk factors and confounders of the logistic regression are the presence of a non-communicable disease (also called the predictors, or explanatory).

2) Meaning of OR:

OR is a measure of association between the presence of NCDs (as exposure or independent variables or risk factors and confounders) and the healthy lifestyle behaviors, perceived mental and physical health during the COVID-19 pandemic (as the outcome variable or dependent variable of this logistic regression). OR=1: NCDS does not affect odds of outcome. OR>1: NCDs associated with higher odds of outcome. OR<1: NCDs associated with lower odds of outcome.Odds Ratio (OR).

Point 2:

  • Level of PA - Level of Physical Activity? "PA" is not explained
  • All P-values should be with the same precision. E.g., 0.78 => 0.780. If p<0.0005, just write 0.000

Response 2:Thank you for your careful and insightful remarks. We’ve explained the abbreviation PA and adjusted P-values according to your suggestions.

Reviewer 3 Report

I appreciate the efforts done by the authors

Author Response

Point 1: I appreciate the efforts done by the authors

Response 1: Thank you very much for accepting our manuscript for publication in the IJERPH. We appreciate your precious time in reviewing our paper and providing valuable comments.It was your valuable and insightful comments that led to possible improvementsin the final version of our manuscript.
